# Content preserving text generation
# with attribute controls

**Lajanugen Logeswaran**[*]    **Honglak Lee**[†]    **Samy Bengio**[†]
[*]University of Michigan, [†]Google Brain
`llajan@umich.edu,{honglak,bengio}@google.com`

## Abstract

In this work, we address the problem of modifying textual attributes of sentences. Given an input sentence and a set of attribute labels, we attempt to generate sentences that are compatible with the conditioning information. To ensure that the model generates content compatible sentences, we introduce a reconstruction loss which interpolates between auto-encoding and back-translation loss components. We propose an adversarial loss to enforce generated samples to be attribute compatible and realistic. Through quantitative, qualitative and human evaluations we demonstrate that our model is capable of generating fluent sentences that better reflect the conditioning information compared to prior methods. We further demonstrate that the model is capable of simultaneously controlling multiple attributes.

## 1   Introduction

Generative modeling of images and text has seen increasing progress over the last few years. Deep generative models such as variational auto-encoders [1], adversarial networks [2] and Pixel Recurrent Neural Nets [3] have driven most of this success in vision. Conditional generative models capable of providing fine-grained control over the attributes of a generated image such as facial attributes [4] and attributes of birds and flowers [5] have been extensively studied. The style transfer problem which aims to change more abstract properties of an image has seen significant advances [6, 7].

The discrete and sequential nature of language makes it difficult to approach language problems in a similar manner. Changing the value of a pixel by a small amount has negligible perceptual effect on an image. However, distortions to text are not imperceptible in a similar way and this has largely prevented the transfer of these methods to text.

In this work we consider a generative model for sentences that is capable of expressing a given sentence in a form that is compatible with a given set of conditioning attributes. Applications of such models include conversational systems [8], paraphrasing [9], machine translation [10], authorship obfuscation [11] and many others. Sequence mapping problems have been addressed successfully with the sequence-to-sequence paradigm [12]. However, this approach requires training pairs of source and target sentences. The lack of parallel data with pairs of similar sentences that differ along certain stylistic dimensions makes this an important and challenging problem.

We focus on categorical attributes of language. Examples of such attributes include sentiment, language complexity, tense, voice, honorifics, mood, etc. Our approach draws inspiration from style transfer methods in the vision and language literature. We enforce content preservation using auto-encoding and back-translation losses. Attribute compatibility and realistic sequence generation are encouraged by an adversarial discriminator. The proposed adversarial discriminator is more data efficient and scales better to multiple attributes with several classes more easily than prior methods.

Evaluating models that address the transfer task is also quite challenging. Previous works have mostly focused on assessing the attribute compatibility of generated sentences. These evaluations do not penalize vacuous mappings that simply generate a sentence of the desired attribute value while

ignoring the content of the input sentence. This calls for new metrics to objectively evaluate models for content preservation. In addition to evaluating attribute compatibility, we consider new metrics for content preservation and generation fluency, and evaluate models using these metrics. We also perform a human evaluation to assess the performance of models along these dimensions.

We also take a step forward and consider a writing style transfer task for which parallel data is available. Evaluating the model on parallel data assesses it in terms of all properties of interest: generating content and attribute compatible, realistic sentences. Finally, we show that the model is able to learn to control multiple attributes simultaneously. To our knowledge, we demonstrate the first instance of learning to modify multiple textual attributes of a given sentence without parallel data.

## 2 Related Work

**Conditional Text Generation**  Prior work have considered controlling aspects of generated sentences in machine translation such as length [13], voice [14], and honorifics/politeness [10]. Kiros et al. [15] use multiplicative interactions between a word embeddings matrix and learnable attribute vectors for attribute conditional language modeling. Radford et al. [16] train a character-level language model on Amazon reviews using LSTMs [17] and discover that the LSTM learns a 'sentiment unit'. By clamping this unit to a fixed value, they are able to generate label conditional paragraphs.

Hu et al. [18] propose a generative model of sentences which can be conditioned on a sentence and attribute labels. The model has a VAE backbone which attempts to express holistic sentence properties in its latent variable. A generator reconstructs the sentence conditioned on the latent variable and the conditioning attribute labels. Discriminators are used to ensure attribute compatibility. Training sequential VAE models has proven to be very challenging [19, 20] because of the posterior collapse problem. Annealing techniques are generally used to address this issue. However, reconstructions from these models tend to differ from the input sentence.

**Style Transfer**  Recent approaches have proposed neural models learned from non-parallel text to address the text style transfer problem. Li et al. [21] propose a simple approach to perform sentiment transfer and generate stylized image captions. Words that capture the stylistic properties of a given sentence are identified and masked out, and the model attempts to reconstruct the sentence using the masked version and its style information. Shen et al. [22] employ adversarial discriminators to match the distribution of decoder hidden state trajectories corresponding to real and synthetic sentences specific to a certain style. Prabhumoye et al. [23] assume that translating a sentence to a different language alters the stylistic properties of a sentence. They adopt an adversarial training approach similar to Shen et al. [22] and replace the input sentence using a back-translated sentence obtained using a machine-translation system.

To encourage generated sentences to match the conditioning stylistic attributes, prior discriminator based approaches train a classifier or adversarial discriminator specific to each attribute or attribute value. In contrast, our proposed adversarial loss involves learning a single discriminator which determines whether a sentence is both realistic and is compatible with a given set of attribute values. We demonstrate that the model can handle multiple attributes simultaneously, while prior work has mostly focused on one or two attributes, which limits their practical applicability.

**Unsupervised Machine Translation**  There is growing interest in discovering latent alignments between text from multiple languages. Back-translation is an idea that is commonly used in this context where mapping from a source domain to a target domain and then mapping it back should produce an identical sentence. He et al. [24] attempt to use monolingual corpora for machine translation. They learn a pair of translation models, one in each direction, and the model is trained via policy gradients using reward signals coming from pre-trained language models and a back-translation constraint. Artetxe et al. [25] proposed a sequence-to-sequence model with a shared encoder, trained using a de-noising auto-encoding objective and an iterative back-translation based training process. Lample et al. [26] adopt a similar approach but with an unshared encoder-decoder pair. In addition to de-noising and back-translation losses, adversarial losses are introduced to learn a shared embedding space, similar to the aligned-autoencoder of Shen et al. [22]. While the auto-encoding loss and back-translation loss have been used to encourage content preservation in prior work, we identify shortcomings with these individual losses: auto-encoding prefers the copy solution and back-translated samples can be noisy or incorrect. We propose a reconstruction loss which interpolates between these two losses to reduce the sensitivity of the model to these issues.

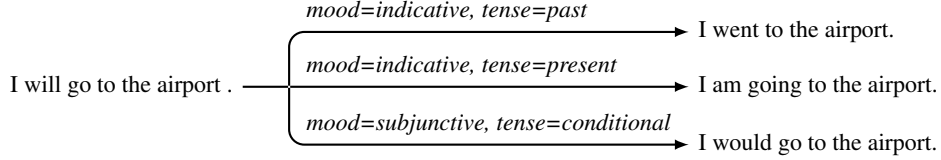

Figure 1: Task formulation - Given an input sentence and attributes values (Eg: indicative mood, past tense) generate a sentence that preserves the content of the input sentence and is compatible with the attribute values.

# 3 Formulation

Suppose we have $K$ attributes of interest $\{a_1, ..., a_K\}$. Let be given a set of labelled sentences $D = \{(x^n, l^n)\}_{n=1}^N$ where $l^n$ is a set of labels for a subset of the attributes. Given a sentence $x$ and attribute values $l' = (l_1, ..., l_K)$ our goal is to produce a sentence that shares the content of $x$, but reflects the attribute values specified by $l'$ (figure 1). In this context, we define *content* as the information in the sentence that is not captured by the attributes. We use the term *attribute vector* to refer to a binary vector representation of the attribute labels. This is a concatenation of one-hot vector representations of the attribute labels.

## 3.1 Model Overview

We denote the generative model by $G$. We want $G$ to use the conditioning information effectively. i.e., $G$ should generate a sentence that is closely related in meaning to the input sentence and is consistent with the attributes. We design $G = (G_{\text{enc}}, G_{\text{dec}})$ as an encoder-decoder model. The encoder is an RNN that takes the words of input sentence $x$ as input and produces a content representation $z_x = G_{\text{enc}}(x)$ of the sentence. Given a set of attribute values $l'$, a decoder RNN generates sequence $y \sim p_G(\cdot|z_x, l')$ conditioned on $z_x$ and $l'$.

## 3.2 Content compatibility

We consider two types of reconstruction losses to encourage content compatibility.

**Autoencoding loss** Let $x$ be a sentence and the corresponding attribute vector be $l$. Let $z_x = G_{\text{enc}}(x)$ be the encoded representation of $x$. Since sentence $x$ should have high probability under $G(\cdot|z_x, l)$, we enforce this constraint using an auto-encoding loss.

$$\mathcal{L}^{ae}(x, l) = -\log p_G(x|z_x, l) \tag{1}$$

**Back-translation loss** Consider $l'$, an arbitrary attribute vector different from $l$ (i.e., corresponds to a different set of attribute values). Let $y \sim p_G(\cdot|z_x, l')$ be a generated sentence conditioned on $x, l'$. Assuming a well-trained model, the sampled sentence $y$ will preserve the content of $x$. In this case, sentence $x$ should have high probability under $p_G(\cdot|z_y, l)$ where $z_y = G_{\text{enc}}(y)$ is the encoded representation of sentence $y$. This requirement can be enforced in a back-translation loss as follows.

$$\mathcal{L}^{bt}(x, l) = -\log p_G(x|z_y, l) \tag{2}$$

A common pitfall of the auto-encoding loss in auto-regressive models is that the model learns to simply copy the input sequence without capturing any informative features in the latent representation. A de-noising formulation is often considered where noise is introduced to the input sequence by deleting, swapping or re-arranging words. On the other hand, the generated sample $y$ can be mismatched in content from $x$ during the early stages of training, so that the back-translation loss can potentially misguide the generator. We address these issues by interpolating the latent representations of ground truth sentence $x$ and generated sentence $y$.

**Interpolated reconstruction loss** We merge the autoencoding and back-translation losses by fusing the two latent representations $z_x, z_y$. We consider $z_{xy} = g \odot z_x + (1 - g) \odot z_y$, where $g$ is a binary random vector of values sampled from a Bernoulli distribution with parameter $\Gamma$. We define a new reconstruction loss which uses $z_{xy}$ to reconstruct the original sentence.

$$\mathcal{L}^{int} = \mathbb{E}_{(x,l)\sim p_{\text{data}}, y\sim p_G(\cdot|z_x, l')}\left[-\log p_G(x|z_{xy}, l)\right] \tag{3}$$

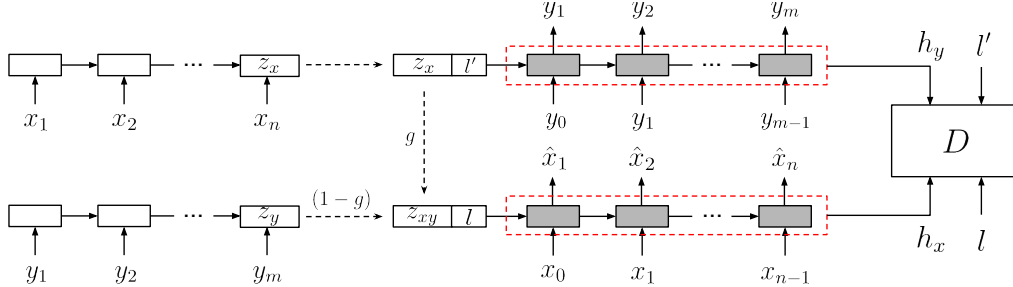

Figure 2: Given an input sentence $x$ with attribute labels $l$, we construct an arbitrary label assignment $l'$ and sample a sentence $y \sim p_G(x, l')$. Given the content representations of $x$ and $y$, an interpolated representation $z_{xy}$ is computed. The decoder reconstructs the input sentence using $z_{xy}$ and $l$. An adversarial discriminator $D$ encourages sequence $y$ to be both realistic and compatible with $l'$.

Note that $\mathcal{L}^{\text{int}}$ degenerates to $\mathcal{L}^{\text{ae}}$ when $g_i = 1$ $\forall i$, and to $\mathcal{L}^{\text{bt}}$ when $g_i = 0$ $\forall i$. The interpolated content embedding makes it harder for the decoder to learn trivial solutions since it cannot rely on the original sentence alone to perform the reconstruction. Furthermore, it also implicitly encourages the content representations $z_x$ and $z_y$ of $x, y$ to be similar, which is a favorable property of the encoder.

## 3.3 Attribute compatibility

We consider an adversarial loss which encourages generating realistic and attribute compatible sentences. The advesarial loss tries to match the distribution of sentence and attribute vector pairs $(s, a)$ where the sentence can either be a real or generated sentence. Let $h_x$ and $h_y$ be the decoder hidden-state sequences corresponding to $x$ and $y$ respectively. We consider an adversarial loss of the following form, where $D$ is a discriminator. Sequence $h_x$ is held constant and $l' \neq l$.

$$\mathcal{L}^{\text{adv}} = \min_G \max_D \mathbb{E}_{(x,l) \sim p_{\text{data}}, y \sim p_G(\cdot | z_x, l')}[\log D(h_x, l) + \log(1 - D(h_y, l'))] \quad (4)$$

It is possible that the discriminator ignores the attributes and makes the real/fake decision based on just the hidden states, or vice versa. To prevent this situation, we consider additional fake pairs $(x, l')$ similar to [5] where we consider a real sentence and a mismatched attribute vector, and encourage the discriminator to classify these pairs as fake. The new objective takes the following form.

$$\mathcal{L}^{\text{adv}} = \min_G \max_D \mathbb{E}_{\substack{(x,l) \sim p_{\text{data}} \\ y \sim p_G(\cdot | z_x, l')}}[2\log D(h_x, l) + \log(1 - D(h_y, l')) + \log(1 - D(h_x, l'))]] \quad (5)$$

Our discriminator architecture follows the projection discriminator [27],

$$D(s, l) = \sigma(l_v^T W \phi(s) + v^T \phi(s)) \quad (6)$$

where $l_v$ represents the binary attribute vector corresponding to $l$. $\phi$ is a bi-directional RNN encoder ($\phi(\cdot)$ represents the final hidden state). $W, v$ are learnable parameters and $\sigma$ is the sigmoid function.

The overall loss function is given by $\mathcal{L}^{\text{int}} + \lambda \mathcal{L}^{\text{adv}}$ where $\lambda$ is a hyperparameter.

## 3.4 Discussion

**Soft-sampling and hard-sampling** A challenging aspect of text generation models is dealing with the discrete nature of language, which makes it difficult to generate a sequence and then obtain a learning signal based on it. Soft-sampling is generally used to back-propagate gradients through the sampling process where an approximation of the sampled word vector at every time-step is used as the input for the next time-step [18, 22]. Inference performs hard-sampling, where sampled words are used instead. Thus, when soft-sampled sequences are used at training time, the training and inference behavior are mismatched. For instance, Shen et al. [22]'s adversarial loss encourages the hidden-state dynamics of teacher-forced and soft-sampled sequences to be similar. However, there remains a gap between the dynamics of these sequences and sequences hard-sampled at test time. We eliminate this gap by hard-sampling the sequence $y$. Soft-sampling also has a tendency to introduce artifacts during generation. These approximations further become poor with large vocabulary sizes. We present an ablative experiment comparing these two sampling strategies in Appendix C.

**Scalability to multiple attributes**   Shen et al. [22] use multiple class-specific discriminators to match the class conditional distributions of sentences. In contrast, our proposed discriminator models the joint distribution of realistic sentences and corresponding attribute labels. Our approach is more data-efficient and exploits the correlation between different attributes as well as attributes values.

## 4    Experiments

The sentiment attribute has been widely considered in previous work [18, 22]. We first address the sentiment control task and perform a comprehensive comparison against previous methods. We perform quantitative, qualitative and human evaluations to compare sentences generated by different models. Next we evaluate the model in a setting where parallel data is available. Finally we consider the more challenging setting of controlling multiple attributes simultaneously and show that our model easily extends to the multiple attribute scenario.

### 4.1    Training and hyperparameters

We use the following validation metrics for model selection. The autoencoding loss $\mathcal{L}^{\text{ae}}$ is used to measure how well the model generates content compatible sentences. Attribute compatibility is measured by generating sentences conditioned a set of labels, and using pre-trained attribute classifiers to measure how well the samples match the conditioning labels.

For all tasks we use a GRU (Gated Recurrent Unit [28]) RNN with hidden state size 500 as the encoder $G_{\text{enc}}$. Attribute labels are represented as a binary vector, and an attribute embedding is constructed via linear projection. The decoder $G_{\text{dec}}$ is initialized using a concatenation of the representation coming from the encoder and the attribute embedding. Attribute embeddings of size 200 and a decoder GRU with hidden state size 700 were used (These parameters are identical to [22]). The discriminator receives an RNN hidden state sequence and an attribute vector as input. The hidden state sequence is encoded using a bi-directional RNN $\phi$ with hidden state size 500. The interpolation probability $\Gamma \in \{0, 0.1, 0.2, .., 1.0\}$ and weight of the adversarial loss $\lambda \in \{0.5, 1.0, 1.5\}$ are chosen based on the validation metrics above. Word embeddings are initialized with pre-trained GloVe embeddings [29].

### 4.2    Metrics

Although the evaluation setups in prior work assess how well the generated sentences match the conditioning labels, they do not assess whether they match the input sentence in content. For most attributes of interest, parallel corpora do not exist. Hence we define objective metrics that evaluate models in a setting where ground truth annotations are unavailable. While individually these metrics have their deficiencies, taken together they are helpful in objectively comparing different models and performing consistent evaluations across different work.

**Attribute accuracy**   To quantitatively evaluate how well the generated samples match the conditioning labels we adopt a protocol similar to [18]. We generate samples from the model and measure label accuracy using a pre-trained sentiment classifier. For the sentiment experiments, the pre-trained classifiers are CNNs trained to perform sentiment analysis at the review level on the Yelp and IMDB datasets [30]. The classifiers achieve test accuracies of 95%, 90% on the respective datasets.

**Content compatibility**   Measuring content preservation using objective metrics is challenging. Fu et al. [31] propose a content preservation metric which extracts features from word embeddings and measures cosine similarity in the feature space. However, it is hard to design an embedding based metric which disregards the attribute information present in the sentence. We take an indirect approach and measure properties that would hold if the models do indeed produce content compatible sentences. We consider a content preservation metric inspired by the unsupervised model selection criteria of Lample et al. [26] to evaluate machine-translation models without parallel data. Given two non-parallel datasets $D_{\text{src}}, D_{\text{tgt}}$ and translation models $M_{src \rightarrow tgt}, M'_{tgt \rightarrow src}$ that map between the two domains, the following metric is defined.

$$f_{\text{content}}(M, M') = 0.5[\mathbb{E}_{x \sim D_{\text{src}}} \text{BLEU}(x, M' \circ M(x)) + \mathbb{E}_{x \sim D_{\text{tgt}}} \text{BLEU}(x, M \circ M'(x))] \quad (7)$$

where $M \circ M'(x)$ represents translating $x \in D_{\text{src}}$ to domain $D_{\text{tgt}}$ and then back to $D_{\text{src}}$. We assume $D_{\text{src}}$ and $D_{\text{tgt}}$ to be test set sentences of positive and negative sentiment respectively and $M, M'$ to be the generative model conditioned on positive and negative sentiment, respectively.

| Model | Yelp Reviews | | | | IMDB Reviews | | | |
|---|---|---|---|---|---|---|---|---|
| | Attribute ↑ | Content ↑ | | Fluency ↓ | Attribute ↑ | Content ↑ | | Fluency ↓ |
| | Accuracy | B-1 | B-4 | Perp. | Accuracy | B-1 | B-4 | Perp. |
| Ctrl-gen [18] | 76.36% | 11.5 | 0.0 | 156 | 76.99% | 15.4 | 0.1 | 94 |
| Cross-align [22] | 90.09% | 41.9 | 3.9 | 180 | 88.68% | 31.1 | 1.1 | 63 |
| Ours | **90.50%** | **53.0** | **7.5** | **133** | **94.46%** | **40.3** | **2.2** | **52** |

Table 1: Quantitative evaluation for sentiment conditioned generation. Attribute compatibility represents label accuracy of generated sentences, as measured by a pre-trained classifier. Content preservation is assessed based on $f_{content}$ (BLEU-1 (B-1) and BLEU-4 (B-4) scores). Fluency is evaluated in terms of perplexity of generated sentences as measured by a pre-trained classifier. Higher numbers are better for accuracy and content compatibility, and lower numbers are better for perplexity.

| Model | Attribute | Content | Fluency |
|---|---|---|---|
| Ctrl-gen [18] | 66.0% | 6.94% | 2.51 |
| Cross-align [22] | 91.2% | 22.04% | 2.54 |
| Ours | **92.8%** | **55.10%** | **3.19** |

Table 2: Human assessment of sentences generated by the models. Attribute and content scores indicate percentage of sentences judged by humans to have the appropriate attribute label and content respectively. Fluency scores were obtained on a 5 point Likert scale.

| Supervision | Model | BLEU |
|---|---|---|
| Paired data | Seq2seq | 10.4 |
| | Seq2seq-bi | 11.15 |
| Unpaired data | Ours | 7.65 |
| Paired + Unpaired data | Ours | **13.89** |

Table 3: Translating Old English to Modern English. The seq2seq models are supervised with parallel data. We consider our model in the unsupervised (no parallel data) and semi-supervised (paired and unpaired data) settings.

**Fluency** We use a pre-trained language model to measure the fluency of generated sentences. The perplexity of generated sentences, as evaluated by the language model, is treated as a measure of fluency. A state-of-the-art language model trained on the Billion words benchmark [32] dataset is used for the evaluation.

### 4.3 Sentiment Experiments

**Data** We use the restaurant reviews dataset from [22]. The dataset is a filtered version of the Yelp reviews dataset. Similar to [18], we use the IMDB move review corpus from [33]. We use Shen et al. [22]'s filtering process to construct a subset of the data for training and testing. The datasets respectively have 447k, 300k training sentences and 128k, 36k test sentences.

We compare our model against Ctrl-gen, the VAE model of Hu et al. [18] and Cross-align, the cross alignment model of Shen et al. [22]. Code obtained from the authors is used to train models on the datasets. We use a pre-trained model provided by [18] for movie review experiments.

**Quantitative evaluation** Table 1 compares our model against prior work in terms of the objective metrics discussed in the previous section. Both [18, 22] perform soft-decoding, so that back-propagation through the sampling process is made possible. But this leads to artifacts in generation, producing low fluency scores. Note that the fluency scores do not represent the perplexity of the generators, but perplexity measured on generated sentences using a pre-trained language model. While the absolute numbers may not be representative of the generation quality, it serves as a useful measure for relative comparison.

We report BLEU-1 and BLEU-4 scores for the content metric. Back-translation has been effectively used for data augmentation in unsupervised translation approaches. The interpolation loss can be thought of as data augmentation in the feature space, taking into account the noisy nature of parallel text produced by the model, and encourages content preservation when modifying attribute properties. The cross-align model performs strongly in terms of attribute accuracy, however it has difficulties generating grammatical text. Our model is able to outperform these methods in terms of all metrics.

**Qualitative evaluation** Table 4 shows samples generated from the models for given conditioning sentence and sentiment label. For each query sentence, we generate a sentence conditioned on the opposite label. The Ctrl-gen model rarely produces content compatible sentences. Cross-align produces relevant sentences, while parts of the sentence are ungrammatical. Our model generates sentences that are more related to the input sentence. More examples can be found in the supplementary material.

| Restaurant reviews | |
|---|---|
| negative → positive | |
| Query | *the people behind the counter were not friendly whatsoever .* |
| Ctrl gen [18] | the food did n't taste as fresh as it could have been either . |
| Cross-align [22] | the owners are the staff is so friendly . |
| Ours | the people at the counter were very friendly and helpful . |
| positive → negative | |
| Query | *they do an exceptional job here , the entire staff is professional and accommodating !* |
| Ctrl gen [18] | very little water just boring ruined ! |
| Cross-align [22] | they do not be back here , the service is so rude and do n't care ! |
| Ours | they do not care about customer service , the staff is rude and unprofessional ! |
| **Movie reviews** | |
| negative → positive | |
| Query | *once again , in this short , there isn't much plot .* |
| Ctrl gen [18] | it's perfectly executed with some idiotically amazing directing . |
| Cross-align [22] | but <unk> , , the film is so good , it is . |
| Ours | first off , in this film , there is nothing more interesting . |
| positive → negative | |
| Query | *that's another interesting aspect about the film .* |
| Ctrl gen [18] | peter was an ordinary guy and had problems we all could <unk> with |
| Cross-align [22] | it's the <unk> and the plot . |
| Ours | there's no redeeming qualities about the film . |

Table 4: Query sentences modified with opposite sentiment by Ctrl gen [18], Cross-align [22] and our model, respectively.

**Human evaluation** We supplement the quantitative and qualitative evaluations with human assessments of generated sentences. Human judges on MTurk were asked to rate the three aspects of generated sentences we are interested in - attribute compatibility, content preservation and fluency. We chose 100 sentences from the test set randomly and generated corresponding sentences with the same content and opposite sentiment. Attribute compatibility is assessed by asking judges to label generated sentences and comparing the opinions with the actual conditioning sentiment label. For content assessment, we ask judges whether the original and generated sentences are related by the desired property (same semantic content and opposite sentiment). Fluency/grammaticality ratings were obtained on a 5-point Likert scale. More details about the evaluation setup are provided in section B of the appendix. Results are presented in Table 2. These ratings are in agreement with the objective evaluations and indicate that samples from our model are more realistic and reflect the conditioning information better than previous methods.

## 4.4 Monolingual Translation

We next consider a style transfer experiment where we attempt to emulate a particular writing style. This has been traditionally formulated as a monolingual translation problem where aligned data from two styles are used to train translation models. We consider English texts written in old English and address the problem of translating between old and modern English. We used a dataset of Shakespeare plays crawled from the web [9]. A subset of the data has alignments between the two writing styles. The aligned data was split as 17k pairs for training and 2k, 1k pairs respectively for development and test. All remaining 80k sentences are considered unpaired.

We consider two sequence to sequence models as baselines. The first one is a simple sequence to sequence model that is trained to translate old to modern English. The second variation learns to translate both ways, where the decoder takes the domain of the target sentence as an additional input. We compare the performance of models in Table 3. In addition to the unsupervised setting which doesn't use any parallel data, we also train our model in the semi-supervised setting. In this setting we first train the model using supervised sequence-to-sequence learning and fine-tune on the unpaired data using our objective. Our version of the model that does not use any aligned data falls short of the supervised models. However, in the semi-supervised setting we observe an improvement of more than 2 BLEU points over the purely supervised baselines. This shows that the model is capable of finding sentence alignments by exploiting the unlabelled data.

| Mood | Tense | Voice | Neg. | john was born in the camp |
|------|-------|-------|------|---------------------------|
| Indicative | Past | Passive | No | john was born in the camp . |
| Indicative | Past | Passive | Yes | john wasn't born in the camp . |
| Indicative | Past | Active | No | john had lived in the camp . |
| Indicative | Past | Active | Yes | john didn't live in the camp . |
| Indicative | Present | Passive | No | john is born in the camp . |
| Indicative | Present | Passive | Yes | john isn't born in the camp . |
| Indicative | Present | Active | No | john has lived in the camp . |
| Indicative | Present | Active | Yes | john doesn't live in the camp . |
| Indicative | Future | Passive | No | john will be born in the camp . |
| Indicative | Future | Passive | Yes | john will not be born in the camp . |
| Indicative | Future | Active | No | john will live in the camp . |
| Indicative | Future | Active | Yes | john will not survive in the camp . |
| Subjunctive | Cond | Passive | No | john could be born in the camp . |
| Subjunctive | Cond | Passive | Yes | john couldn't live in the camp . |
| Subjunctive | Cond | Active | No | john could live in the camp . |
| Subjunctive | Cond | Active | Yes | john couldn't live in the camp . |

Table 5: Simultaneous control of multiple attributes. Generated sentences for all valid combinations of the input attribute values.

## 4.5 Ablative study

Figure 3 shows an ablative study of the different loss components of the model. Each point in the plots represents the performance of a model (on the validation set) during training, where we plot the attribute compatibility against content compatibility. As training progresses, models move to the right. Models at the top right are desirable (high attribute and content compatibility). $\mathcal{L}^{ae}$ and $\mathcal{L}^{int}$ refer to models trained with only the auto-encoding loss or the interpolated loss respectively. We observe that the interpolated reconstruction loss by itself produces a reasonable model. It augments the data with generated samples and acts as a regularizer. Integrating the adversarial loss $\mathcal{L}^{adv}$ to each of the above losses improves the attribute compatibility since it explicitly requires generated sequences to be label compatible (and realistic). We also consider $\mathcal{L}^{ae} + \mathcal{L}^{bt} + \mathcal{L}^{adv}$ in our control experiment. While this model performs strongly, it suffers from the issues associated with $\mathcal{L}^{ae}$ and $\mathcal{L}^{bt}$ discussed in section 3.2. The attribute compatibility of the proposed model $\mathcal{L}^{int} + \mathcal{L}^{adv}$ drops more gracefully compared to the other settings as the content preservation improves.

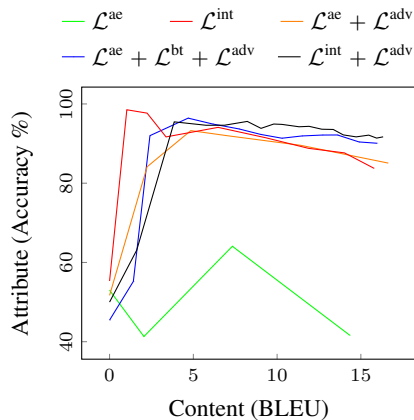

Figure 3: For different objective functions, we plot attribute compatibility against content compatibility of the learned model as training progresses. Models at the top right are desirable (High compatibility along both dimensions).

## 4.6 Simultaneous control of multiple attributes

In this section we discuss experiments on simultaneously controlling multiple attributes of the input sentence. Given a set of sentences annotated with multiple attributes, our goal is to be able to plug this data into the learning algorithm and obtain a model capable of tweaking these properties of a sentence. Towards this end, we consider the following four attributes: tense, voice, mood and negation. We use an annotation tool [34] to annotate a large corpus of sentences. We do not make fine distinctions such as progressive and perfect tenses and collapse them into a single category. We used a subset of ~2M sentences from the BookCorpus dataset [15], chosen to have approximately near class balance across different attributes.

Table 5 shows generated sentences conditioned on all valid combinations of attribute values for a given query sentence. We use the annotation tool to assess attribute compatibility of generated sentences. Attribute accuracies measured on generated senetences for mood, tense, voice, negation were respectively 98%, 98%, 90%, 97%. The voice attribute is more difficult to control compared to

the other attributes since some sentences require global changes such as switching the subject-verb-object order, and we found that the model tends to distort the content during voice control.

## 5  Conclusion

In this work we considered the problem of modifying textual attributes in sentences. We proposed a model that explicitly encourages content preservation, attribute compatibility and generating realistic sequences through carefully designed reconstruction and adversarial losses. We demonstrate that our model effectively reflects the conditioning information through various experiments and metrics. While previous work has been centered around controlling a single attribute and transferring between two styles, the proposed model easily extends to the multiple attribute scenario. It would be interesting future work to consider attributes with continuous values in this framework and a much larger set of semantic and syntactic attributes.

**Acknowledgements** We thank Andrew Dai, Quoc Le, Xinchen Yan and Ruben Villegas for helpful discussions. We also thank Jongwook Choi, Junhyuk Oh, Kibok Lee, Seunghoon Hong, Sungryull Sohn, Yijie Guo, Yunseok Jang and Yuting Zhang for helpful feedback on the manuscript.

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
