[Supplementary Material]

# Supplementary Material: Content preserving text generation with attribute controls

**Lajanugen Logeswaran*** **Honglak Lee**[†] **Samy Bengio**[†]
*University of Michigan, [†]Google Brain
llajan@umich.edu,{honglak,bengio}@google.com

## A Qualitative comparison

| Restaurant Reviews | |
|---|---|
| *negative → positive* | |
| Query | *sorry but i do n't get the rave reviews for this place .* |
| Ctrl gen | i ordered the nachos , have perfect seasonal beans on amazing amazing . |
| Cross-align | sorry , i do n't be the best experience ever . |
| Ours | thanks but i love this place for lunch . |
| Query | *however my recent visit there made me change my mind entirely .* |
| Ctrl-gen | not like other target stores . |
| Cross-align | best little one time to go for in charlotte . |
| Ours | overall my experience here was great as well . |
| Query | *okay so this place has been a pain even after i already moved out .* |
| Ctrl gen | like i mentioned , i thought to this these fun . |
| Cross-align | food and this place has been a good place to be back . |
| Ours | overall this is a great place to go when i 'm in town . |
| Query | *personally i 'd rather spend my money at a business that appreciates my business .* |
| Ctrl gen | i became quite a gem at the beginning but we amazing fantastic amazing . |
| Cross align | then i will be back my time to get a regular time . |
| Ours | definitely i 'll definitely be back for a good haircut . |
| *positive → negative* | |
| Query | *best chinese food i 've had in a long time .* |
| Ctrl gen | very lousy texture and ruined . |
| Cross align | worst chinese food i 've had in a long in years . |
| Ours | worst food i 've had in a long time . |
| Query | *my appetizer was also very good and unique .* |
| Ctrl gen | both were ruined . ruined |
| Cross align | my wife was just very bland and no flavor . |
| Ours | my chicken was very dry and had no flavor . |
| Query | *everything tasted great and the service was excellent .* |
| Ctrl gen | but the real pleasure is the service department . |
| Cross align | everything tasted horrible and the service was very bad . |
| Ours | everything tasted bad and the service was horrible . |
| Query | *atmosphere is cozy and comfortable .* |
| Ctrl gen | atmosphere is not good . |
| Cross align | rude is dirty and way in . |
| Ours | restaurant is dirty and dirty . |

Table 1: Query sentences from the restaurant reviews test dataset modified with opposite sentiment by Ctrl gen [1], Cross-align [2] and our model, respectively.

| **Movie Reviews** | |
|---|---|
| negative → positive | |
| Query | *this is the most vapid movie i have ever seen .* |
| Ctrl gen | if this grabs your interest , you may want to give it a try |
| Cross-align | this is a great movie that is so good . |
| Ours | this is the most beautiful movie i have ever seen . |
| Query | *this 1944 film is too awful as it 's just incredible .* |
| Ctrl gen | <unk> the three dead world and <unk> 's <unk> is a cult in a life |
| Cross-align | this film is one of the best movies ever made . |
| Ours | this film is an excellent and it is definitely worth it . |
| Query | *1 out of 10 .* |
| Ctrl gen | he 's cold and hateful exactly what his part <unk> |
| Cross-align | my rating of the cast . |
| Ours | 10 out of 10 . |
| Query | *i always thought she was a colorless , plain jane .* |
| Ctrl gen | a great comedy all wrapped up in a tiny package ! |
| Cross-align | i think that is the best of the film . |
| Ours | i also thought she was a beautiful , talented actor . |
| Query | *her character is truly hateful and her acting , if you can call it that , is strictly wretched .* |
| Ctrl gen | a great ' proper ' summer movie |
| Cross-align | <unk> , is the <unk> , and you can be able to be more than it to be . |
| Ours | his character is very funny , and in fact , it 's just what he does n't disappoint . |
| positive → negative | |
| Query | *this is one of his best efforts .* |
| Ctrl gen | as david <unk> picked up the franchise , it has just <unk> to pieces |
| Cross-align | this is a complete waste of time . |
| Ours | this is one of the worst films . |
| Query | *if you love silent films , you 'll adore this one .* |
| Ctrl gen | nice photographic effects as jessica <unk> the process |
| Cross-align | if you 're no , but it is not bad . |
| Ours | if you love horror movies , do n't see this one . |
| Query | *and congratulations to kino for a superb video restoration .* |
| Ctrl gen | peter <unk> is not that she 's not gone bad movie |
| Cross-align | but then , it 's a waste of time . |
| Ours | and save your money on this piece of garbage . |
| Query | *the characters are portrayed vividly and realistically .* |
| Ctrl gen | problem is , not enough good work went into this |
| Cross-align | the characters are <unk> and <unk> . |
| Ours | the characters are completely unsympathetic and annoying . |
| Query | *there are some of the most stunning and grisly combat scenes ever filmed .* |
| Ctrl gen | unfortunately the only thing you see is <unk> |
| Cross-align | there is no a <unk> , and the <unk> , <unk> and <unk> . |
| Ours | there are some of the most boring and boring scenes ever made . |

Table 2: Query sentences from the movie reviews test dataset modified with opposite sentiment by Ctrl gen [1], Cross-align [2] and our model, respectively.

# B  Human Evaluation

Sections B.1, B.2, B.3 describe the setup for human evaluations done using Amazon Mechanical Turk (AMT) to obtain annotations for generated sentences.

## B.1  Content compatibility

Given a reference sentence and a set of candidate sentences, pick the candidates that have the **same semantic content as the reference sentence but have the opposite sentiment** (i.e., mean the opposite). Select all that apply. If you think neither of the given sentences have this property, choose *No preference* (This can happen when all the candidate sentences are either semantically irrelevant to the reference sentence or have the incorrect sentiment).

Example:
Reference sentence: *This is a great movie !*
You would pick sentences such as
   ✓ *This is not a good movie.*
   ✓ *This is a bad movie.*
The following sentences do not fit the criteria because they are either semantically irrelevant to the reference sentence or have the incorrect sentiment.
   ✗ *I did not like the salad.*
   ✗ *This is a wonderful movie.*

## B.2  Attribute compatibility

Pick the best sentiment based on the following criterion.

| Sentiment | Guidance |
|---|---|
| Positive | Sentence conveys positive sentiment. Eg: "I really liked the food." |
| Negative | Sentence conveys negative sentiment. Eg: "This was the worst experience ever." |
| Neutral | Sentence does not carry any sentiment information. |

## B.3  Fluency

Rate the grammaticality/fluency of the sentence based on the following criterion.

| Fluency | Guidance |
|---|---|
| 5 | The sentence is grammatical and does not have any grammar errors. |
| 4 | Sentence is mostly grammatical except for one/two mistakes. |
| 3 | Parts of the sentence are grammatical and sentence is somewhat coherent, but there are glaring errors. |
| 2 | Too many grammatical errors and sentence is incoherent. |
| 1 | Sentence is completely ungrammatical. |

## C Sampling strategy

In this section we compare soft and hard sampling during training. For the soft-sampling model, we use an exponential decay temperature annealing schedule with an initial temperature of 1. The temperature decays until it reaches 0.01 and remains constant afterwards. Other parameters of the model are identical to section 4.1 of the paper. We use the Yelp dataset for this experiment. Table 3 compares the models with respect to the metrics in section 4.2.

Models learned with soft-sampling produce sentences judged to be highly attribute compatible. However, the content compatibility is considerably poor and generated sentences have lower fluency. This supports our claim that the training and inference behavior are mismatched when soft-sampled sequences are used for training.

| Model | Attribute ↑ compatibility | Content ↑ compatibility | | Fluency ↓ |
|---|---|---|---|---|
| | | BLEU-1 | BLEU-4 | Perplexity |
| Hard-sampling | 90.50% | **53.0** | **7.5** | **133** |
| Soft-sampling | **92.33**% | 43.6 | 3.1 | 137 |

Table 3: Comparison between soft and hard sampling. Higher numbers are better for accuracy and content compatibility and lower numbers are better for perplexity.