[Reviews · NeurIPS 2018]

Reviewer 1



This paper proposed generating text that reflects a desired set of attributes while preserving content by using an interpolated loss to promote content preservation and an adversarial loss to adhere to the desired attributes. They evaluate the results with both quantitatively and with human evaluations, principally along three qualities: attribute accuracy, content preservation, and fluency. This is a challenging task to both execute and evaluate, and I appreciated the authors' focus on both pieces. Generally, the paper was clear and addressed an interesting problem. Section 3 could use more clarity about which pieces are novel, with citations to support the pieces that are not. Some questions: -I was excited for the multiple attribute aspect, but the verb task wasn't what I was expecting. It seems like most of these attributes can be injected into a sentence in a very templated way. Could you motivate why you chose this task as opposed to extending the sentiment task with other attributes (or even adding the mood/tense/voice/negation attributes to the sentiment task)? -On a related note, how much diversity is there in the way that the attributes are incorporated in the generated sentences? -How did you decide how to set the interpolation probability between auto-encoding and back-translation losses? Minor comments: -The caption in Table 4 says query sentences are indicated by color, but the table is in black and white. -"english" should be capitalized, particularly noticeable in section 4.4

Reviewer 2



The authors aim to achieve style transfer in text governed by controllable attributes. For this, authors propose a model which builds on prior works on unsupervised machine translation and style transfer. The proposed model uses reconstruction and back-translation losses. Authors propose to add denoising via dropping inputs and using an interpolated hidden representation. An additional adversarial loss is added to ensure the generated distribution of (hx,l) matches the input data distribution. An undesirable solution for D can be to simply use hx and hy, and ignore the attributes. To deal with this, (hx,l') is added as additional negative sample. Authors test their proposed model and training methods on multiple datasets - sentiment change, Shakespearean English style transfer, active/passive/tense/etc. on Book Corpus dataset. Human evaluation is carried out in some cases. An ablative study is also carried out to highlight the importance of different terms in the loss function. While the proposed model and experiments seem interesting, there seems to be limited novelty in the used model compared to prior works. Description of model and loss functions is not clear and seems incomplete at times. For example, in section 3.3, it seems that authors want to use adversarial loss to distinguish b/w 'real' and 'generated/fake' sentence-attribute tuples - shouldn't then the equation 4 use max over D and min over G? Additionally, is there anything preventing decoder output states hx to encode information about attributes l without actually using it? If hx does include information about l, then isn't the task of D much easier? In such a situation, role of D is not clear. I would suggest authors to add more discussions and explanations for their model choices. Most of the experiments seem convincing - the proposed method performs pretty well on a variety of datasets. 4.6 describes an important set of experiments wherein multiple attributes are changed - however this part misses on many details such as distribution of attribute values and number of sentences in data, and performance wrt content preservation. Discussions and analysis about the types of errors the model makes on the three datasets could be added for better understanding of the working of the model. I am interested in understanding more about instructions provided for human evaluation, especially with respect to content preservation. Authors point out that annotators were asked 'whether the original and generated sentences are related by the desired property' - what is 'property' here? Is it same as attribute? In that case, this seems a bit ill-defined since 'related' is a very loose relation - can two wildly different sentences with negative sentiments can be said to be related wrt the sentiment attribute? How was the meaning of the term 'related' conveyed to the annotators? Authors could have provided more details or a screenshot of evaluation framework in the supplementary material. Update: I have considered the authors' response, and have increased my score to 7.

Reviewer 3



Thanks for the response. The added discussions make sense to me so I've increased my rating. It'd be necessary to include these discussions in the reviesed verison. ------- Original reviews: This paper develops a new model for unsupervised text style transfer, which includes components that encourage content compatibility and attribute compatibility. Experiments show improved performance. * The task of text style transfer has two well-adopted goals, namely “content compatibility" and "attribute compatibility”, using the terminology from the paper. For these goals, many techniques/components have been invented in the literature. It is easy to pick an arbitrary subset of these components and compose a "new" model. It is thus critical to make clear what difference (and advantages) the particular components and the particular combination used in the proposed model have. However, - I didn’t see many such discussions when the authors present the model. The discussion in section 3.4 instead focuses on the sampling strategy which seems relatively minor. It’s necessary to provide more discussions. For example, back-translation is used in [22] as well; Adversarial loss on decoder hidden state is also used in [21]. What is the difference of the respective components used in the new model? Try to make it clearer. - Some ablation study is conducted trying to validate the advantages of some of the specifications. For example, the effect of the auto-encoding loss and the interpolated loss. But I’m wondering what is the performance of using only the back-translation loss? And, what is the performance of simply combining both the auto-encoding loss and the back-translation loss with fixed weight (without annealing)? Such specifications are simpler than the interpolated loss. Does the interpolation really necessary? - Similarly, if there is any difference between the adversarial loss and that of [21], is it possible to conduct ablation study to see how the new adversarial loss compared to that of [21]? * The paper claims this is the first work to modify multiple attributes without parallel data. However, [18] has also simultaneously controlled more than one attributes with non-parallel and separate data. * Section 3.4 makes a discussion of soft-sampling and hard-sampling, and claims hard-sampling is better. But there is no ablation study to support the claim. Questions: * It looks the results (e.g., attribute accuracy) on the Yelp Reviews data differ greatly from those reported in [21]. Is there any difference regarding the experimental settings?